# Preeclampsia: Maternal Systemic Vascular Disorder Caused by Generalized Endothelial Dysfunction Due to Placental Antiangiogenic Factors

**DOI:** 10.3390/ijms20174246

**Published:** 2019-08-30

**Authors:** Takuji Tomimatsu, Kazuya Mimura, Shinya Matsuzaki, Masayuki Endo, Keiichi Kumasawa, Tadashi Kimura

**Affiliations:** 1Department of Obstetrics and Gynecology, Osaka University Graduate School of Medicine, Osaka 565-0871, Japan; 2Department of Obstetrics and Gynecology, Tokyo University Graduate School of Medicine, Tokyo 113-0033, Japan

**Keywords:** preeclampsia, endothelial dysfunction, angiogenic imbalance, systemic vascular dysfunction, arterial stiffness, vascular endothelial growth factor, placental growth factor, soluble fms-like tyrosine kinase 1

## Abstract

Preeclampsia, a systemic vascular disorder characterized by new-onset hypertension and proteinuria after 20 weeks of gestation, is the leading cause of maternal and perinatal morbidity and mortality. Maternal endothelial dysfunction caused by placental factors has long been accepted with respect to the pathophysiology of preeclampsia. Over the past decade, increased production of placental antiangiogenic factors has been identified as a placental factor leading to maternal endothelial dysfunction and systemic vascular dysfunction. This review summarizes the recent advances in understanding the molecular mechanisms of endothelial dysfunction caused by placental antiangiogenic factors, and the novel clinical strategies based on these discoveries.

## 1. Introduction

Preeclampsia, a systemic vascular disorder of pregnancy characterized by hypertension in association with proteinuria, affects 5% to 10% of all pregnancies. This condition can affect virtually every organ system, causing preeclampsia-related adverse complications such as seizures (eclampsia), HELLP (hemolysis, elevated liver enzymes, and low platelets) syndrome, abruptio placentae, and fetal growth restriction. Currently, the only effective treatment is termination of pregnancy, which may also add substantial risks to the neonate if the fetus is delivered prematurely. Although the clinical symptoms of preeclampsia completely resolve after delivery, recent evidence has demonstrated significant association between history of preeclampsia and future risks of cardiovascular diseases [1,2].

Generalized maternal endothelial dysfunction due to placental factors has been considered to play an important role in the pathogenesis of preeclampsia. Many serum markers indicating endothelial activation increase [3], and flow-mediated dilation (FMD), the gold standard for evaluating endothelial function [4], is impaired in patients with preeclampsia [5]. Over the past decade, excess placental antiangiogenic factor, soluble fms-like tyrosine kinase 1(sFlt1), has been shown to antagonize vascular endothelial growth factor (VEGF) and placental growth factor (PlGF), and to induce generalized endothelial dysfunction in these women [6,7,8,9]. This discovery generated great enthusiasm for sFlt1 as the most promising placental factor and led to the development of novel clinical strategies for managing preeclampsia. In this review, we aimed to summarize the recent advances in understanding the pathophysiology of preeclampsia, and novel clinical strategies from the viewpoints of maternal endothelial dysfunction caused by placental antiangiogenic factors.

## 2. Diagnosis and Risk Factors of Preeclampsia

Preeclampsia is defined as an increase in systolic blood pressure ≥140/90 mmHg in previously normotensive women as well as proteinuria ≥300 mg in a 24-h collection, or 0.3 g/g by urine protein/creatinine ratio, or +1 by urine dipstick if it is the sole available test, occurring after 20 weeks of gestation. These diagnostic criteria do not require proteinuria for diagnosis in the presence of other organ damage such as thrombocytopenia, renal failure, liver involvement, cerebral symptoms, and pulmonary edema [10].

Several risk factors for preeclampsia clearly indicate the presence of genetic predisposition. The incidence of preeclampsia is greatly affected by race and ethnicity. African-American women have a higher risk of developing preeclampsia than white women [11]. Population based cohort studies revealed that the daughters and sons of women who had preeclampsia during pregnancy had a higher risk of developing preeclampsia themselves and fathering a preeclamptic pregnancy, respectively [12,13]. Recently, a large clinical genome-wide association study revealed significant association between single-nucleotide polymorphism near the FLT1 locus (rs4769613) on chromosome 13 in the fetal genome and the development of preeclampsia [14,15]. Trisomy 13 (chromosome 13 contains the FLT1 locus) is associated with increased maternal levels of sFlt1 and a high risk of preeclampsia [16].

Multifetal [17] and molar pregnancies [18] are associated with an increased risk of preeclampsia, presumably due to increased levels of sFlt1. We have reported an interesting case of mirror syndrome (maternal preeclampsia-like symptoms and fetal hydrops) caused by severe fetal anemia due to parvovirus B19 infection. Both maternal and fetal symptoms resolved immediately after intrauterine transfusion, along with the normalization of the increased levels of sFlt1 [19]. These findings may strengthen the importance of the central contribution of maternal antiangiogenic state in the pathogenesis of preeclampsia.

Reduced paternal antigen exposure such as nulliparity, shorter periods of sexual cohabitation, and changing paternity demonstrate increased risks for developing preeclampsia, indicating immunological contribution to the pathogenesis [20,21]. Some maternal conditions, such as advanced maternal age, obesity, diabetes mellitus, chronic hypertension, antiphospholipid syndrome (APS), chronic kidney disease (CKD), and systemic lupus erythematosus (SLE), are also associated with an increased risk of preeclampsia [1,2]. These maternal conditions have been reported to be associated with endothelial dysfunction, which may contribute to increased risk of preeclampsia [22]. In addition, it has been reported that pregnant women with chronic hypertension and pregnant women with diabetes showed significantly higher sflt1 level, and pregnant women with obesity had significantly lower PlGF level [17].

## 3. Maternal Antiangiogenic State in Preeclampsia

Maternal endothelial dysfunction caused by placental factors and the two-stage theory have long been accepted with regard to the pathophysiology of preeclampsia (Figure 1) [23]. The initial step is considered to start from insufficient cytotrophoblast invasion of spiral arteries (abnormal placentation) [24,25,26,27]. Predisposing immunological, genetic, and preexisting maternal risk factors may affect this abnormal placentation. In preeclampsia, failure of the physiological remodeling of decidual vessels results in reduced placental perfusion [28], which has been believed to release placental factors into the maternal circulation.

In 2003, Maynard et al. [6] discovered increased levels of placental sFlt1 in the serum of women with preeclampsia. They also reported that administration of sFlt1 to pregnant rats generated preeclampsia-like symptoms. Subsequently, Levine et al. [7] showed that serum levels of sFlt1 correlates with disease severity and declines after resolution. These experimental and epidemiological studies as well as several following landmark reports [8,9], generated compelling evidence that placental sFlt1 is one of the most important placental factors leading to maternal endothelial dysfunction [29].

sFlt1 is a splice variant of the VEGFR1 (Flt1) containing only the extracellular ligand-binding domain of VEGFR1 [30]. As VEGFR1 (Flt1) interacts with VEGF and PlGF [31], sFlt1 inhibits proangiogenic signaling by antagonizing VEGF and PlGF. Importantly, sFlt1 is mainly produced by the trophoblastic cells, and released into the maternal circulation during pregnancy [32] (Figure 2).

## 4. Mechanisms of Endothelial Dysfunction by Inhibiting the VEGF Signal Pathway

The precise mechanisms of endothelial dysfunction in women with preeclampsia due to high levels of circulating sFlt1 remains unclear. Direct administration of VEGF augments the release of nitric oxide (NO) from the vascular endothelium [33] and causes nitric oxide-dependent hypotension in vivo [34]. VEGF has been shown to stimulate NO production via upregulation of nitric oxide synthase (NOS) expression in endothelial cells [35,36]. This vasodilation effect of VEGF may be mediated by both VEGFR1 (Flt1) and VEGFR2 (KDR/FlK1) receptors, but VEGFR2 is the predominant receptor mediating this effect [37]. VEGF, but not PlGF, was also shown to induce prostacyclin (PGI_2_) synthesis [38,39]. NO activates soluble guanylate cyclase (sGC), leading to cGMP synthesis. PGI_2_ activates adenylyl cyclase (AC) and increases cAMP synthesis. Both cGMP and cAMP lead to decreased intracellular Ca^2+^ concentrations, which induce smooth muscle relaxation and vasodilation [40].

The recent introduction of VEGF inhibitor therapies in cancer patients and its preeclampsia-like adverse effects, particularly hypertension and renal injury, have attracted much attention and made doubly sure that inhibiting the VEGF signal pathway is central to the pathophysiology in preeclampsia [41,42,43,44,45,46]. VEGF inhibitor therapies in cancer patients also added several interesting insights into the mechanisms of preeclampsia. Of these, several lines of evidence reported a dose-dependent activation of the endothelin-1 (ET-1) in response to VEGF inhibitor therapies [47,48,49]. Considering that ET-1 is the most potent vasoconstrictor and increased levels of ET-1 have also been reported in women with preeclampsia [50,51], it seems plausible that ET-1 is involved in pathogenesis. Although the precise mechanism leading to increased levels of ET-1 by inhibiting the VEGF signal pathway remains unclear, it has been reported that VEGF enhances prepro-ET-1 mRNA expression and induces endothelin-converting enzyme-1 (ECE-1), which is a key enzyme in endothelin processing [52,53].

## 5. Mechanism of Renal Injury by Inhibiting the VEGF Signal Pathway

VEGF is synthesized by podocytes within the glomerulus where it maintains fenestrated endothelium [54]. Inhibiting the VEGF signal pathway causes endothelial swelling, termed glomerular endotheliosis, which is the renal lesion frequently seen in women with preeclampsia [55,56]. The importance of NO in maintaining normal renal function has also been well documented [57]. Inhibiting the VEGF signal pathway reduces NO production due to the decreasing expression of endothelial and neuronal NOS in the kidney [58]. In addition, it has been reported that proteinuria and glomerular endotheliosis caused by VEGF inhibitor therapies were prevented by the endothelin receptor blocker [59], indicating the involvement of the endothelin system in the mechanism of renal injury by inhibiting the VEGF signal pathway.

## 6. Complement System and Angiogenic Imbalance

There has been compelling evidence that complement activation is implicated in the pathogenesis of preeclampsia [60,61]. Clinical similarities between atypical hemolytic uremic syndrome (aHUS), a disease of excessive activation of the alternative complement pathway, and HELLP syndrome, a severe variant of preeclampsia, along with the findings of several studies [62,63], have strengthened the role of complement activation in the pathogenesis of preeclampsia. Elevated levels of urinary C5b-9 in women with preeclampsia have been shown to be a useful biomarker that differentiates preeclampsia from other hypertensive disorders [64,65]. Immunohistochemical studies using renal biopsy specimens from women with preeclampsia revealed increased renal C4d-a and C1q-positive glomeruli, suggesting the importance of the classical complement pathway in the pathogenesis [66]. Increased C4 deposits in the glomeruli were also shown in the sFlt1-injected pregnant mouse model, indicating that angiogenic dysregulation may play a role in complement activation within the kidney [66]. It has also been reported that aHUS was also induced in cancer patients under VEGF inhibitor therapies [67]. Recently, a possible mechanism linking the complement system to angiogenic imbalance was shown, in which inhibiting the VEGF signal pathway decreases local complement inhibitor synthesis in renal glomeruli, potentially making these sites vulnerable to complement activation [68]. Another study reported that human extravillous trophoblast cell line HTR-8/Svneo treated with C5a expressed significantly increased mRNA levels of sFlt1 and decreased mRNA levels of PlGF [69].

## 7. Regulating sFlt1 Production by Trophoblastic Cells

The mechanisms of placental sFlt1 upregulation are largely unknown. Alternative splicing of the pre-mRNA encoding FLT1 results in the production of sFlt1 containing only the extracellular ligand-binding domain of Flt1 but lacking the intracellular and membrane-spanning domains [30]. It is believed that hypoxic environment caused by abnormal placentation stimulates sFlt1 production [70]. In accordance with this notion, elevated expression of transcription factor hypoxia-inducible factor 1α (HIF1α) was shown to contribute to sFlt1 upregulation in in vivo and in vitro models of human placenta [71]. Inhibition of complement activation has been shown to block the increase of sFlt1 in pregnant mice [72]. Mitochondrial dysfunction leading to reactive oxygen species generation and oxidative stress may contribute to sFlt1 production [73]. Recently, the upregulation of VEGF in maternal decidual cell was advocated as a trigger of sFlt1 production by trophoblastic cells [74].

## 8. Preeclampsia as a Systemic Vascular Disorder of Pregnancy

Systemic vascular dysfunction is considered as a final step in the pathophysiology of preeclampsia (Figure 1). During normal pregnancy, maternal vascular resistance decreases, resulting in slightly decreased blood pressure [75,76,77,78,79,80]. In women with preeclampsia, it has been thought that these adaptations do not occur sufficiently due to systemic vascular disorder with generalized endothelial dysfunction. Although the precise mechanism of systemic vascular disorder caused by endothelial dysfunction remains elusive, abnormalities in matrix metalloproteinases (MMPs) and increased collagen deposition in extracellular matrix (ECM) are considered to play significant roles in inadequate vascular remodeling leading to systemic vascular dysfunction [81].

Recently, noninvasive assessment of vascular function [82] has directly revealed the presence of systemic vascular dysfunction in women with preeclampsia. FMD has been shown to increase during pregnancy [83]. In women with preeclampsia, significantly lower FMD was found both before and after the development of the disease as well as 3 years after delivery [84]. VEGF inhibitor therapies with bevacizumab have also been shown to result in reduced endothelium-mediated vasodilation [45]. Pulse wave analysis (PWA) measures the composite stiffness of the conduit and resistance artery [85,86,87]. PWA indices (augmentation index and central systolic pressure) decline markedly during pregnancy [88,89,90]. In women with preeclampsia, these indices are significantly increased [91,92]. Abnormal PWA measures have also been observed before the onset of disease [93,94], as well as 6–24 months postpartum [95]. Increased PWA measures were also shown to be more relevant to intrauterine fetal growth than conventional brachial blood pressure [96,97]. Pulse wave velocity (PWV) has been considered to provide information regarding the stiffness of conduit arteries [98,99], and is elevated in pregnant women with preeclampsia [100,101].

The results from these vascular function tests provided a comprehensive picture of a systemic vascular dysfunction due to endothelial dysfunction, the final step in the pathophysiology of preeclampsia (Figure 1). It is expected that combining vascular function tests with the assessment of angiogenic imbalance might improve prediction accuracy of preeclampsia-related adverse complications. In addition, these tests revealed the presence of vascular dysfunction even after the resolution of clinical symptoms of preeclampsia, indicating its possible association with future cardiovascular disease risks.

## 9. Novel Clinical and Therapeutic Strategies from the Viewpoints of Maternal Angiogenic Imbalance

### 9.1. Prediction of Disease

Identifying women at risk for preeclampsia in the first trimester is now an area of important clinical research, as low-dose aspirin started before 16 weeks of gestation was reported to be associated with a significant decrease in the prevalence of preeclampsia in women identified to be at high-risk based on the conventional maternal clinical risk factors, such as nulliparity, a history of preeclampsia, and chronic hypertension [102].

Although extensive investigations revealed that the value of sFlt1 levels in the first trimester showed no clear association with the development of preeclampsia [103], PlGF levels in the first trimester have been shown to have consistent and promising results in the prediction of preeclampsia [104]. Recently, screening performance for preeclampsia in first trimester based on an algorithm combining PlGF levels with maternal clinical factors, mean arterial pressure (MAP), and uterine artery pulsatility index (UtA-PI) was reported to be by far superior to the screening performance based on conventional maternal clinical risk factors [105]. Using this algorithm, a recent large clinical trial (ASPRE trial) confirmed that first-trimester screening combining with low-dose aspirin administration resulted in a substantial decrease in the incidence of preterm preeclampsia (odds ratio, 0.38; 95% confidence interval (CI), 0.20–0.74; *p* = 0.004) [106].

### 9.2. Prediction of Adverse Maternal and Perinatal Complications

Clinically, it is not unusual for pregnant women without hypertension or without proteinuria to develop preeclampsia-related adverse complications, such as eclampsia [107,108,109]. Conversely, a significant portion of women who meet the diagnostic criteria for preeclampsia do not show any adverse complications and are able to carry a pregnancy to nearly full term. Therefore, the association of maternal angiogenic imbalance with the occurrence of preeclampsia-related adverse complications, rather than with the development of preeclampsia, has been vigorously investigated.

In pregnant women with suspected preeclampsia, the severity of the maternal antiangiogenic state predicted preeclampsia-related adverse complications more accurately than the highest systolic blood pressure, a hallmark of the diagnostic criteria for preeclampsia [110,111]. In women with suspected preeclampsia presenting at <34 weeks, an sFlt1/PlGF ratio ≥85 predicted preterm delivery within 2 weeks with a hazard ratio of 15.2 [110]. Furthermore, a secondary analysis of this study revealed that patients who meet diagnostic criteria for preeclampsia but had a normal angiogenic profile showed no preeclampsia-related adverse maternal and fetal complications [112]. In women with suspected preeclampsia presenting at ≤36 weeks, an sFlt-1/PlGF ratio <38 showed an extremely high negative predictive value of 99.3% on the occurrence of preeclampsia-related adverse complications within 1 week [113]. Furthermore, a secondary analysis of this study revealed that patients with an sFlt1/PlGF ratio ≥38 showed significantly shorter remaining time to delivery and a higher rate of preterm delivery, irrespective of the development of preeclampsia [114]. In 90% of women with suspected or confirmed preeclampsia with an sFlt1/PlGF ratio ≤38, the ratio was largely stable and did not increase further up to 100 days [115]. Recently, a randomized controlled trial confirmed that implementing PlGF measurement in managing women with suspected preeclampsia significantly improved maternal outcome [116].

### 9.3. Assessing Angiogenic Imbalance in Differential Diagnosis

Assessing angiogenic imbalance is reported to be useful in differentiating preeclampsia from other diseases with preeclampsia-like symptoms, including chronic kidney disease (CKD) [117], gestational thrombocytopenia [118], and chronic hypertension [119]. In case of a flare of SLE during pregnancy, assessing angiogenic imbalance can lead to appropriate treatment (prednisolone escalation), instead of unnecessary iatrogenic preterm deliveries [120,121]. In pregnant women with SLE, APS, or both, circulating angiogenic factors measured during early gestation have a high negative predictive value of 93% in ruling out the development of severe adverse outcomes, including early-onset preeclampsia, fetal/neonatal death, and iatrogenic preterm delivery before 30 weeks of gestation [122].

### 9.4. Therapeutic Potential of Modulating Angiogenic Factors

A large number of basic research studies have suggested the therapeutic potential of modulating angiogenic factors [123,124,125,126,127,128,129,130], including administration of recombinant VEGF [126] or PlGF [127], and reducing sFlt1 levels via RNA interference (RNAi) [130]. However, currently, only one pilot human trial aimed at direct modulation of maternal angiogenic imbalance has reported a clinical benefit, in which removal of sFlt1 by dextran sulfate apheresis resulted in the stabilization of blood pressure and prolongation of pregnancy in women with very preterm (<32 weeks) preeclampsia [131].

## 10. Prevention of Preeclampsia

### 10.1. Low-Dose Aspirin

The role of aspirin in the primary or secondary prevention of preeclampsia has long been an important clinical concern. In preeclampsia, thromboxane A2 (TXA_2_: platelet activator and vasoconstrictor) production by the platelets increases, whereas PGI_2_ production by the endothelium decreased [132]. Both TXA_2_ and PGI_2_ are synthesized from arachidonic acid by the action of cyclooxygenase (COX) [133]. Although low-dose acetylsalicylic acid (aspirin) blocks COX irreversibly, the endothelium recovers PGI_2_ production by de novo synthesis of COX [134]. However, the platelets, where TXA_2_ is synthesized, cannot synthesize COX as the platelets are anuclear. In accordance with this notion, several studies reported that low-dose aspirin reduced TXA_2_ production without altering the PGI_2_ production [135,136], leading to normal TXA_2_/PGI_2_ balance by two weeks of treatment [137]. Aspirin is also shown to have angiogenic properties by blocking sFlt1 production in human trophoblasts [138] and by increasing PlGF production in BeWo trophoblast cells [139].

Although the first clinical trials showed significant efficacy of aspirin in preventing preeclampsia in 1985 [140], subsequent large-scale clinical studies have shown limited or no clinical benefit of low-dose aspirin for the prevention of preeclampsia [141,142,143]. In 2010, Bujold et al. published a meta-analysis of double-blind randomized trials and suggested a greater benefit when aspirin treatment was started before 16 weeks of gestation (relative risk (RR), 0.47; 95% CI 0.34–0.65) in high-risk patients [102]. As mentioned above, a recent multicenter, double blind, randomized, placebo-controlled trial (ASPRE trial) evaluated the effect of low-dose aspirin, administered from 11 to 14 weeks of gestation until 36 weeks of gestation, among women who were identified as high risk following first-trimester screening. This trial revealed a substantial reduction in the incidence of preterm preeclampsia (odds ratio, 0.38; 95% CI, 0.20–0.74; *p* = 0.004) [106].

### 10.2. Low-Dose Aspirin plus Heparin

Antiphospholipid syndrome (APS) is an autoimmune disease that causes an increased risk of thrombotic or adverse obstetrical events in patients with persistent antiphospholipid antibodies [144]. In pregnant women with APS, of whom a third of these women develop preeclampsia, treatment with low-dose aspirin plus heparin has long been the most efficacious regimen, with significant improvement of maternal and perinatal outcomes [144]. In a mouse model of APS, one of the beneficial mechanisms of heparin in women with APS was shown to be mediated by inhibition of the complement system [145], which also can be beneficial in women with preeclampsia.

For preventing preeclampsia in patients with and without APS, a recent meta-analysis reported that heparin improved the efficacy of low-dose aspirin alone (RR, 0.54; 95% CI 0.31–0.92) [146]. However, subsequent large multicenter trials did not show the potential efficacy of heparin for the prevention of preeclampsia in high-risk patients without APS, indicating that heparin may benefit only a subset of patients [147,148]. Moreover, several studies reported the contradictory but consistent observations that heparin increases both circulating PlGF and sFlt1 levels, suggesting that heparin does not have ideal properties for restoring angiogenic imbalance observed in women with preeclampsia [149,150,151].

## 11. Preeclampsia and Future Risk of Cardiovascular Disease

As mentioned above, it is now established that preeclampsia is associated with the future risks for cardiovascular diseases [152,153,154,155]. Women with a history of preeclampsia are at increased risks for future cardiovascular diseases, such as chronic hypertension (RR, 3.7; 95% CI, 2.70–5.05) [153], and heart failure [RR, 4.19 (2.09–8.38)] [152], stroke (RR, 1.81 (1.29–2.55)) [152], coronary heart disease (RR, 2.50 (1.43–4.37)) [152], and cardiovascular mortality (RR, 2.21 (1.83–2.66)) [152] when compared to women without a history of preeclampsia.

Although the precise mechanisms remain unclear, several recent studies revealed that the persistence of maternal endothelial and vascular dysfunction after delivery by using several non-invasive vascular function tests or echocardiography [95,156,157]. Although the sFlt1 level has been reported to decrease to less than 1% of its pre-delivery value within 24 h of the delivery [115], the levels of sFlt1 and the sFlt1/PlGF ratio were still higher at 1 year postpartum in women with preeclampsia [158]. Elevated levels of sFlt1 were also reported at 5–8 years postpartum in women with preeclampsia [159].

Recently, the clinical significance of sFlt1 in cardiac functions has been advocated. Increased levels of sFlt1 were related to the development of acute heart failure in patients with acute myocardial infarction [160]. In women with preeclampsia, the extent of subclinical cardiac dysfunction correlates with the circulating levels of sFlt1, and women with peripartum cardiomyopathy showed abnormally increased level of sFlt1 even 4–6 weeks postpartum [161]. In addition, systemic administration of sFlt1 in a mouse model of peripartum cardiomyopathy induced substantial cardiac dysfunction [161].

Although a history of preeclampsia is now recognized as a women-specific risk factor for cardiovascular disease in later life, it is still unclear how the cardiovascular health of these women should be improved. Presently, several guidelines suggest management for monitoring of hypertension, hyperlipidemia, and diabetes, and provision of healthy lifestyle advice for women with a history of preeclampsia [162]. Further studies are needed to define the appropriate monitoring and intervention strategies for these women.

## 12. Conclusions

Our understanding of the pathophysiology of preeclampsia has advanced considerably in the past decade. Maternal systemic vascular dysfunction caused by generalized endothelial dysfunction due to placental antiangiogenic factors has emerged as one of the most important mechanisms, and clinical strategies from the viewpoints of maternal angiogenic imbalance has been increasingly incorporated in the clinical practice. In the future, novel clinical and therapeutic strategies aimed at restoring angiogenic imbalance is expected to ameliorate complications and prolong gestation in women with preeclampsia. In addition, elucidation of pathophysiology and establishment of effective screening and prevention strategies will guide clinicians to reduce future risks of cardiovascular disease in women with preeclampsia.

## Figures and Tables

**Figure 1 ijms-20-04246-f001:**
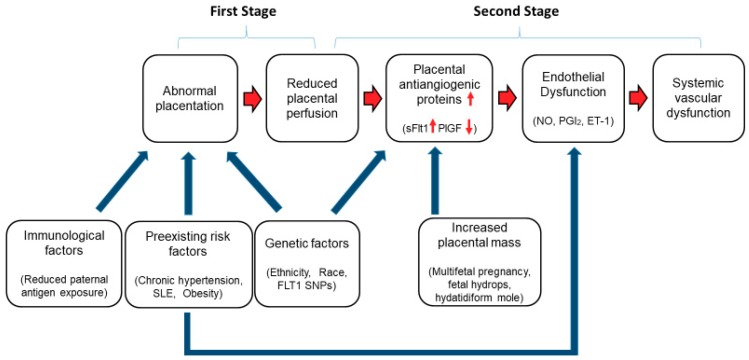
Two-stage theory of the pathophysiology of preeclampsia. Predisposing immunological, genetic, and preexisting maternal risk factors may affect abnormal cytotrophoblast invasion of spiral arteries (abnormal placentation) (First stage). The reduced uteroplacental perfusion induces placental release of antiangiogenic factors (soluble fms-like tyrosine kinase 1 (sFlt1)) into the maternal circulation, which antagonizes proangiogenic factors, leading to endothelial dysfunction and systemic vascular dysfunction (Second stage). Preexisting maternal conditions such as chronic hypertension, systemic lupus erythematosus (SLE), and obesity is also associated with endothelial dysfunction.

**Figure 2 ijms-20-04246-f002:**
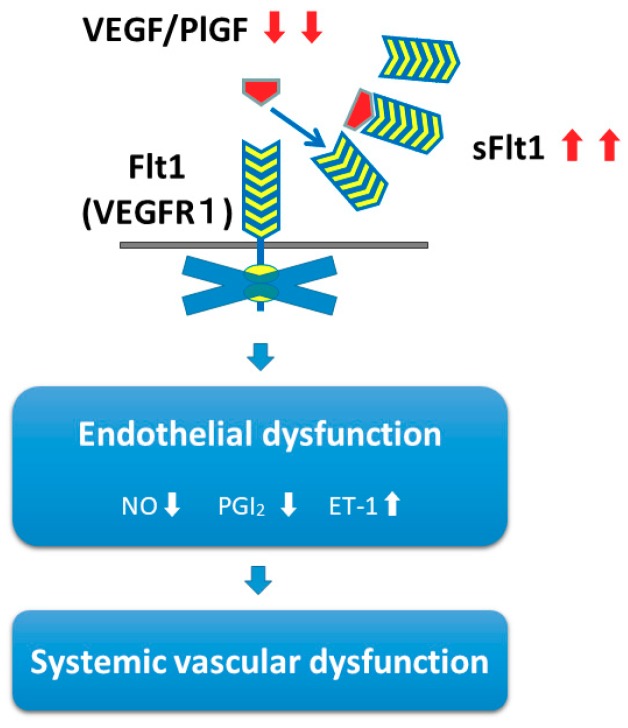
Mechanisms of endothelial dysfunction leading to systemic vascular dysfunction in preeclampsia. Excessive soluble fms-like tyrosine kinase 1 (sFlt1) antagonizes VEGF or PlGF, or both, and causes endothelial dysfunction, including a decrease in vasodilators such as nitric oxide (NO) and prostacyclin (PGI_2_) and an increase in vasoconstrictors such as endothelin-1 (ET-1). VEGF: vascular endothelial growth factor; PLGF: placental growth factor; VEGFR1: VEGF receptor 1 (also known as Flt1).

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
