# Peer review of "Preeclampsia: Maternal Systemic Vascular Disorder Caused by Generalized Endothelial Dysfunction Due to Placental Antiangiogenic Factors"

_ijms, 2019, doi:10.3390/ijms20174246_

Round 1

Reviewer 1 Report

A good and comprehensive review of antiangiogenic factors involved in preeclampsia with a good introduction to possible therapies based on them. The review is very good, I would only address the need of a more in-depth analysis of the picture surrounding preeclampsia (mainly obesity and diabetes and hypertension) which I think should broad the readers interested in the review.

Author Response

Response to Reviewer 1 Comments

Point 1: I A good and comprehensive review of antiangiogenic factors involved in preeclampsia with a good introduction to possible therapies based on them. The review is very good, I would only address the need of a more in-depth analysis of the picture surrounding preeclampsia (mainly obesity and diabetes and hypertension) which I think should broad the readers interested in the review.

Response 1: 

We appreciate the helpful suggestions from the referee, as the comments were very useful for revising this manuscript.  

Along with reviewer’s suggestion, we added following sentences and references to the revised manuscript.

“These maternal conditions have been reported to be associated with endothelial dysfunction, which may contribute to increased risk of preeclampsia [22]. In addition, it has been reported that pregnant women with chronic hypertension and pregnant women with diabetes showed significantly higher sflt1 level, and pregnant women with obesity had significantly lower PlGF level [23].”

Zhang, H.N.; Xu, Q.Q.; Thakur, A.; Alfred, M.O.; Chakraborty, M.; Ghosh, A.; Yu, X.B. Endothelial dysfunction in diabetes and hypertension: Role of microRNAs and long non-coding RNAs. Life Sci. 2018, 213, 258-268.

Maynard, S.E.; Crawford, S.L.; Bathgate, S.; Yan, J.; Robidoux, L.; Moore, M.; Moore Simas, T.A. Gestational angiogenic biomarker patterns in high risk preeclampsia groups. Am J Obstet Gynecol. 2013, 209, 53.e1-9.

In addition, we revised Figure 1 and added following sentence to the figure legend in order to clarify the role of preexisting risk factors such as diabetes, chronic hypertension, and obesity.

“Preexisting maternal conditions such as chronic hypertension, SLE, and obesity is also associated with endothelial dysfunction.”

Reviewer 2 Report

This is a very good review of our current understanding of preeclampsia. It is well written and comprehensive. My only comment/request is that more be included on the production of sFLT. The alternative forms of FLT arise due to alternative RNA transcripts being produced. This needs to be discussed. There is evidence that this might be linked to oxidative stress and mitochondrial function, this deserves further consideration. 

Author Response

Response to Reviewer 1 Comments

Point 1: This is a very good review of our current understanding of preeclampsia. It is well written and comprehensive. My only comment/request is that more be included on the production of sFLT. The alternative forms of FLT arise due to alternative RNA transcripts being produced. This needs to be discussed. There is evidence that this might be linked to oxidative stress and mitochondrial function, this deserves further consideration..

Response 1: 

We appreciate the helpful suggestions from the referee, as the comments were very useful for revising this manuscript.  

Along with reviewer’s suggestion, we added following sentences and reference to the revised manuscript.

"Alternative splicing of the pre-mRNA encoding FLT1 results in the production of sFlt1 containing only the extracellular ligand-binding domain of Flt1 but lacking the intracellular and membrane-spanning domains [31]." 

"Mitochondrial dysfunction leading to reactive oxygen species generation and oxidative stress may contribute to sFlt1 production [74]."  

74. Vaka, V.R.; McMaster, K.M.; Cunningham, MW, Jr.; Ibrahim, T.; Hazlewood, R.; Usry, N.; Cornelius, D.C.; Amaral, L.M.; LaMarca, B. Hypertension. 2018, 72, 703-711.